# SketchDNN: Joint Continuous-Discrete Diffusion for CAD Sketch Generation

**Sathvik Chereddy** [1]    **John Femiani** [1]

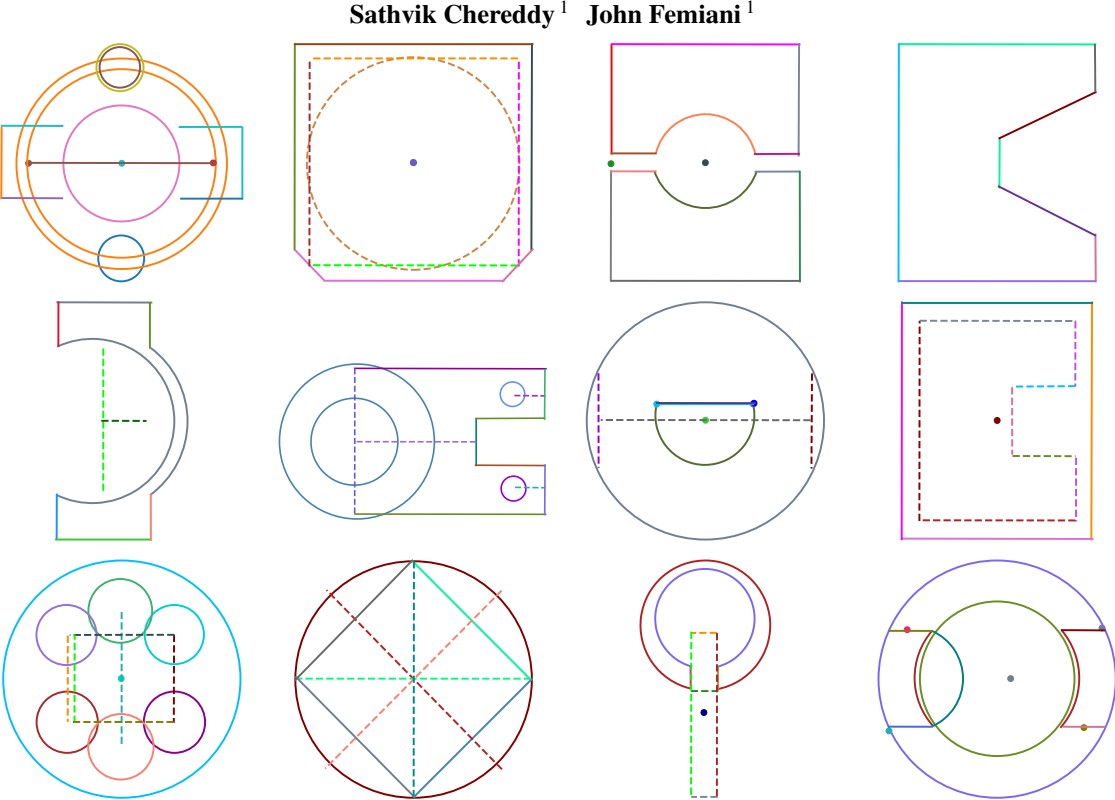

Figure 1. CAD sketches generated by our diffusion model, showcasing its ability to produce diverse, high-fidelity designs. Geometric primitives—such as circles, arcs, lines, and points—are randomly colored to differentiate separate primitives, and primitives tagged as construction aids are represented with dashed lines.

## Abstract

We present SketchDNN, a generative model for synthesizing CAD sketches that jointly models both continuous parameters and discrete class labels through a unified continuous-discrete diffusion process. Our core innovation is Gaussian-Softmax diffusion, where logits perturbed with Gaussian noise are projected onto the probability

simplex via a softmax transformation, facilitating blended class labels for discrete variables. This formulation addresses 2 key challenges, namely, the heterogeneity of primitive parameterizations and the permutation invariance of primitives in CAD sketches. Our approach significantly improves generation quality, reducing Fréchet Inception Distance (FID) from 16.04 to 7.80 and negative log-likelihood (NLL) from 84.8 to 81.33, establishing a new state-of-the-art in CAD sketch generation on the SketchGraphs dataset.

[1]Department of Computer Science, Miami-Oxford University, Oxford OH, USA. Correspondence to: Sathvik Chereddy (M.S.) <sathware@outlook.com>, John Femiani <femianjc@miamioh.edu>.

*Proceedings of the $42^{nd}$ International Conference on Machine Learning*, Vancouver, Canada. PMLR 267, 2025. Copyright 2025 by the author(s).

## 1. Introduction

Computer-aided design (CAD) modeling is a cornerstone of modern engineering that is heavily reliant on human inge-

nuity. The typical workflow in CAD modeling starts with designing 2D planar sketches comprised of primitives (e.g., lines, arcs) that define geometry and constraints (e.g., co-incidence, orthogonality) that define relationships between primitives. Such diagrams are then expanded into 3D volumes using operations such as extrusion and revolution. Over the course of the CAD modeling process, these 3D volumes are iteratively aggregated to form the final 3D model. As a result, sketch construction takes a substantial portion of user's time and effort in CAD design.

A number of researchers have identified CAD diagrams as an ideal domain for generative models (Seff et al., 2020; Ganin et al., 2021; Para et al., 2024; Willis et al., 2021; Wu et al., 2021; Seff et al., 2022). Primitive generation is of particular interest, as constraints do not define geometry but instead serve as editing aids for users. Analogous to image generative models such as Stable Diffusion, CAD generative models can streamline design and improve user productivity (Seff et al., 2020). By automating low-level CAD drawing construction, these models could allow designers to focus more on higher-level design tasks (Ganin et al., 2021). Furthermore, they could facilitate novel workflows (Para et al., 2024), such as simultaneously exploring multiple sketch variations to meet design requirements.

However, generating CAD sketches is difficult due to two fundamental challenges. The first challenge arises from the heterogeneity of sketch primitives, as each primitive type is defined by its own distinct parameterization. The second challenge lies in the permutation invariance of primitives, where any ordering of primitives encodes the same geometry. This poses a particular challenge for autoregressive models, the dominant generation paradigm in this domain, as they generate primitives sequentially. While prior approaches have made significant progress, they do not adequately address these challenges, which we believe limits the diversity and quality of their generated sketches. Diffusion models, which have demonstrated remarkable success in image generation, are a promising avenue for addressing these challenges.

In this work, we introduce a novel generative diffusion model for unconditional primitive generation and propose a discrete diffusion strategy built on the Gaussian-Softmax distribution. Our method addresses both challenges in CAD sketch generation by employing superposition, wherein each primitive is represented as a probabilistic mixture of all primitive types, and permutation-invariant denoising, where the denoising of each primitive is independent of its ordering. Our approach improves the state-of-the-art (SOA) in terms of both generation quality and diversity.

Our key contributions are as follows:

1. We propose the first sketch-space/data-space generative

diffusion model for parametric CAD sketches.

2. We introduce a Gaussian-Softmax based diffusion paradigm for modeling discrete variables.

3. We advance the state of the art in CAD sketch generation. Our model reduces the Fréchet Inception Distance (FID) from 16.04 to 7.80 and the Negative Log-Likelihood (NLL) from 84.8 to 81.33 on the Sketch-Graphs dataset, demonstrating significant improvements in both fidelity and diversity.

## 2. CAD Sketches

A CAD sketch is naturally represented as a graph $\mathcal{G} = (V, E)$, where the set of nodes $V = \{v_i\}$ corresponds to geometric primitives, and the set of edges $E = \{e_{ij}\}$ encodes constraints between them. While constraints play a crucial role in defining design intent, they introduce additional dependencies that significantly complicate generative modeling. In this work, we focus on synthesizing the geometric primitives alone, deferring constraints to future work. Some primitives may be designated as construction aids; such primitives are not directly rendered in the final 3D CAD model but assist in specifying more complex design requirements (dashed lines in Fig. 1).

Each primitive $v \in V$ is characterized by three attributes: (1) a boolean $b$ indicating whether it is a construction aid, (2) a class label $c \in \mathcal{C}$ specifying its type where $\mathcal{C} = \{\textsc{Line}, \textsc{Circle}, \textsc{Arc}, \textsc{Point}\}$, and (3) a set of parameters $p \in \mathbb{R}^{d_c}$, where $d_c$ depends on the primitive class. The parameters used to define each primitive type is as follows:

$$
\begin{aligned}
\textsc{Circle}: \quad & (x, y, r) & \text{(center coords, radius)} \\
\textsc{Line}: \quad & (x_1, y_1, x_2, y_2) & \text{(start/end coords)} \\
\textsc{Arc}: \quad & (x_1, y_1, x_2, y_2, \kappa) & \text{(start/end coords, radius)} \\
\textsc{Point}: \quad & (x, y) & \text{(xy-coordinates)}
\end{aligned}
$$

We note that $\kappa$ is the curvature or signed radius, negating the curvature will reflect the ARC across the line defined by its starting and ending terminal points.

We resolve the heterogeneity of primitive parameterizations by modeling each primitive as a composite structure that encodes all possible primitive types. Specifically, we represent each primitive as:

$$(\mathbf{b}, \mathbf{c}, \mathbf{p}_{\textsc{Line}}, \mathbf{p}_{\textsc{Circle}}, \mathbf{p}_{\textsc{Arc}}, \mathbf{p}_{\textsc{Point}})$$

where $\mathbf{b}$ is a one-hot encoding of the construction aid flag, $\mathbf{c} \in \{0, 1\}^{|\mathcal{C}|+1}$ is a one-hot encoding of the class label with an additional NONE type, and $\mathbf{p}_{\textsc{Line}}, \mathbf{p}_{\textsc{Circle}}, \mathbf{p}_{\textsc{Arc}}, \mathbf{p}_{\textsc{Point}}$ are the parameters for each primitive type. This representation enables SketchDNN to view each primitive as a probabilistic mixture, or superposition, of all primitive types.

Thus, the full sketch representation is given by a matrix $X \in \mathbb{R}^{n \times d}$, where $n$ is the maximum number of primitives, and $d$ is the feature dimension. For this work $n = 16$ and $d = 20$. To recover the standard representation from our composite encoding, we extract the class label with the highest confidence (i.e., $c = \text{argmax}\,\{\mathbf{c}\}$) along with its corresponding parameters.

## 3. Methodology

In this work, we depart from the conventional diffusion parameterization of predicting noise, and instead parameterize our model to predict the ground truth datapoint directly. We observed in our preliminary experiments that this approach yielded higher fidelity samples for CAD diagram generation than predicting the noise added. This finding parallels the results of (Ho et al., 2020), who demonstrated that predicting noise led to superior sample quality in the context of image generation, and so it appears the converse is true for CAD sketch generation. For completeness, and because the foundational principles of diffusion models extend naturally to our proposed Guassian-Softmax diffusion paradigm, we include a concise review of Gaussian diffusion formulated in terms of the ground truth parameterization.

### 3.1. Continuous Diffusion

Diffusion models, first introduced by (Sohl-Dickstein et al., 2015) and later popularized by (Ho et al., 2020), represent a class of generative models that learn to undo a gradual noising process. The forward noising process is a Markov chain that gradually adds Gaussian noise to a clean datapoint $x_0$ over a series of $T$ timesteps, so that $\mathbf{x}_T$ is indistinguishable from pure noise. The forward transition is defined as:

$$\mathbf{x}_t = \sqrt{\alpha_t}\mathbf{x}_{t-1} + \sqrt{(1 - \alpha_t)}\boldsymbol{\epsilon} \tag{1}$$

where $\boldsymbol{\epsilon} \sim \mathcal{N}(0, \mathbf{I})$ and $\alpha_0, \alpha_1, \ldots, \alpha_T$ is a noise schedule that controls the amount of noise added at each timestep. A noise schedule is a sequence of monotonically decreasing scalar values satisfying $\alpha_0 = 1$ and $\alpha_T = 0$. We use the cosine variance schedule introduced by Nichol & Dhariwal (Nichol & Dhariwal, 2021) for this work.

Conveniently, iterative compositions of the forward transition can be expressed in closed form as the cumulative transition:

$$\mathbf{x}_t = \sqrt{\overline{\alpha_t}}\mathbf{x}_0 + \sqrt{(1 - \overline{\alpha_t})}\boldsymbol{\epsilon} \tag{2}$$

where $\overline{\alpha_t} = \prod_{i=1}^{t} \alpha_i$. This allows for efficient training, as we can in one shot sample $\mathbf{x}_t$ from the original datapoint $\mathbf{x}_0$ for any arbitrary $t \in \{0, \ldots, T\}$. The reverse transition is then given by:

$$\mathbf{x}_{t-1} = \boldsymbol{\mu}_{t-1} + \sigma_{t-1}\boldsymbol{\epsilon} \tag{3}$$

where

$$\boldsymbol{\mu}_{t-1} = \frac{\sqrt{\alpha_t}\,(1 - \overline{\alpha}_{t-1})\,\mathbf{x}_t + \sqrt{\overline{\alpha}_{t-1}}\,(1 - \alpha_t)\,\mathbf{x}_0}{1 - \overline{\alpha}_t}$$

and

$$\sigma_{t-1} = \sqrt{\frac{(1 - \alpha_t)\,(1 - \overline{\alpha}_{t-1})}{1 - \overline{\alpha}_t}}$$

We note that $\boldsymbol{\mu}_{t-1}$ is simply a function of $\mathbf{x}_t$ and $\mathbf{x}_0$, so in more explicit terms $\boldsymbol{\mu}_{t-1} := \boldsymbol{\mu}_{t-1}(\mathbf{x}_t, \mathbf{x}_0)$. The reverse process is learned by training a denoiser network to predict the clean data point given a noisy sample. The denoiser network can be expressed as $\hat{\mathbf{x}}_0^\theta(\mathbf{x}_t, t)$, and the learned reverse transition is given by:

$$\mathbf{x}_{t-1} = \boldsymbol{\mu}_{t-1}(\mathbf{x}_t, \hat{\mathbf{x}}_0^\theta(\mathbf{x}_t, t)) + \sigma_{t-1}\boldsymbol{\epsilon}$$

where novel samples are generated from pure noise $\mathbf{x}_T$ by iteratively applying the learned reverse transition from $t = T$ to $t = 0$.

### 3.2. Discrete Diffusion

Building on the success of continuous diffusion models in domains such as image generation, Hoogeboom et al. (2021) introduced a discrete diffusion framework utilizing the Categorical distribution, that was later expanded by (Austin et al., 2021). In Multinomial diffusion, the forward process is a discrete Markov chain that jumbles the state of a one-hot vector $\mathbf{y}_0$ over $T$ steps. The forward transition is defined to be

$$\mathbf{y}_t \sim \text{Cat}(\mathbf{y}_{t-1}Q_t)$$

where $Q_t$ is a probability matrix. In other words, the forward transition stochastically permutes/shuffles the index of the 1 in a one-hot vector. As a result, the output of the forward transition is always a one-hot vector, and similarly the cumulative transition and reverse transition also output one-hot vectors. This naturally renders superposition impossible in conventional discrete diffusion, because there is no way to accommodate blended class labels to express uncertainty. This is detrimental because there is no gradual destruction of information, once a class label is shuffled all information is destroyed. Preliminary experiments with generating CAD sketches using Multionomial diffusion yielded poor results, we believe the root cause to be the inability to accommodate superposition. This is validated by our results in Section 6.

### 3.3. Gaussian-Softmax Diffusion

To overcome the limitations of Multinomial diffusion, we introduce the Gaussian-Softmax ($\mathcal{GS}$) distribution as a continuous relaxation of the Categorical distribution. We propose a novel simplex-constrained discrete diffusion framework utilizing the Gaussian-Softmax distribution to enable

superposition, allowing information to be gradually destroyed in the forward process and gradually recovered in the reverse process. Essentially, if $\mathbf{x} \sim \mathcal{N}(\boldsymbol{\mu}, \sigma^2 \mathbf{I})$, then $\boldsymbol{\sigma}_{\mathrm{SM}}(\mathbf{x}) \sim \mathcal{GS}(\boldsymbol{\mu}, \sigma^2 \mathbf{I})$ where $\boldsymbol{\sigma}_{\mathrm{SM}}$ is the softmax transformation. The softmax function transforms arbitrary vectors $\mathbf{x}$ into probability vectors $\mathbf{p}$, satisfying $\sum_i \mathbf{p}_i = 1$ and $\mathbf{p}_i \geq 0, \forall i$, which geometrically means that the softmax operation maps vectors in $\mathbb{R}^D$ onto the probability simplex $\Delta^{D-1}$. Thus the support of the Gaussian-Softmax distribution is the probability simplex, and the density (derived in Appendix A.2) is:

$$p(\mathbf{y}|\boldsymbol{\mu}, \sigma^2 \mathbf{I}) = \frac{\prod_{i=1}^{D} \mathbf{y}_i}{Z(\sigma)} \exp\left(-\frac{1}{2\sigma^2}\|\tilde{\mathbf{y}} - \boldsymbol{\mu}'\|_\perp^2\right) \quad (4)$$

where

$$Z(\sigma) = \sqrt{D(2\pi\sigma^2)^{(D-1)}}$$

$$\|\tilde{\mathbf{y}} - \boldsymbol{\mu}'\|_\perp^2 = \|\tilde{\mathbf{y}} - \boldsymbol{\mu}'\|^2 - \frac{1}{D}(\mathbf{1}^T(\tilde{\mathbf{y}} - \boldsymbol{\mu}'))^2$$

Here, $\boldsymbol{\mu}' = \boldsymbol{\mu} - (\boldsymbol{\mu}_D)\mathbf{1}$, where every element in $\boldsymbol{\mu}$ is shifted by its last element, and similarly for $\tilde{\mathbf{y}} = \log \mathbf{y} - (\log \mathbf{y}_D)\mathbf{1}$, every element in $\log \mathbf{y}$ is shifted by its last element.

### 3.3.1. FORWARD PROCESS

Analogous to continuous diffusion, the forward process is a Markov chain that progressively noises a one-hot vector $\mathbf{y}_0$ via $\mathcal{GS}$ noise over $T$ steps, generating a sequence of increasingly noisy probability vectors $\mathbf{y}_1, \mathbf{y}_2, \ldots, \mathbf{y}_T$. As noise accumulates, information about the true label is destroyed, and by $t = T$, the class label becomes fully randomized where $\mathbf{y}_T \sim \mathcal{GS}(0, \mathbf{I})$. To achieve this, we define the forward transition as

$$\mathbf{y}_{t+1} = \boldsymbol{\sigma}_{\mathrm{SM}}\left(\sqrt{\alpha_{t+1}} \log \mathbf{y}_t + \sqrt{1 - \alpha_{t+1}}\boldsymbol{\epsilon}\right) \quad (5)$$

where $\boldsymbol{\epsilon} \sim \mathcal{N}(\mathbf{0}, \mathbf{I})$. Additionally, we can conveniently sample $\mathbf{y}_t$ directly from the one-hot vector $\mathbf{y}_0$ using the cumulative transition (derivation in Appendix A.1):

$$\mathbf{y}_t = \boldsymbol{\sigma}_{\mathrm{SM}}\left(\sqrt{\overline{\alpha}_t} \log \mathbf{y}_0' + \sqrt{1 - \overline{\alpha}_t}\boldsymbol{\epsilon}\right) \quad (6)$$

here $\mathbf{y}_0' = k\mathbf{y}_0 + \frac{1-k}{D}\mathbf{1}$ and $k$ is an user-defined constant close to 1, we set $k = 0.99$. We slightly label smooth $\mathbf{x}_0$ to avoid singularities from computing the logarithm of 0. We remark that at $t = T$ the cumulative transition simply takes the softmax of an i.i.d standard Gaussian vector, which by symmetry makes the argmax or label $c_T$ follow a uniform distribution, satisfying $c_T \sim \mathcal{C}\left(\frac{1}{D}\right)$.

### 3.3.2. REVERSE PROCESS

Notably, we find that the discrete reverse transition is analogous to the continuous case and takes the form:

$$\mathbf{y}_{t-1} = \boldsymbol{\sigma}_{\mathrm{SM}}\left(\boldsymbol{\mu}_{t-1}(\mathbf{y}_t, \mathbf{y}_0) + \sigma_{t-1}\boldsymbol{\epsilon}\right) \quad (7)$$

Here, the mean $\boldsymbol{\mu}_{t-1}$ is simply an interpolation between the logits of $\mathbf{y}_0$ and the logits of $\mathbf{y}_t$:

$$\boldsymbol{\mu}_{t-1}(\mathbf{y}_t, \mathbf{y}_0) = \frac{\sqrt{\alpha_t}(1 - \overline{\alpha}_{t-1})\mathbf{y}_t' + \sqrt{\overline{\alpha}_{t-1}}(1 - \alpha_t)\mathbf{y}_0'}{1 - \overline{\alpha}_t}$$

where $\mathbf{y}_t' = \log \mathbf{y}_t$ and $\mathbf{y}_0' = \log \mathbf{y}_0$. Again, The standard deviation is similarly given as:

$$\sigma_{t-1} = \sqrt{\frac{(1 - \alpha_t)(1 - \overline{\alpha}_{t-1})}{1 - \overline{\alpha}_t}}$$

A proof of the reverse process is provided in Appendix A.3.

Intuitively, we perform continuous diffusion in log-space and project back onto the probability simplex using the softmax function. The reverse transition is learned by training a denoiser network to predict $\hat{\mathbf{y}}_0^\theta(\mathbf{y}_t, t)$ where:

$$\mathbf{y}_{t-1} = \boldsymbol{\sigma}_{\mathrm{SM}}\left(\boldsymbol{\mu}_{t-1}\left(\mathbf{y}_t, \hat{\mathbf{y}}_0^\theta(\mathbf{y}_t, t)\right) + \sigma_{t-1}\boldsymbol{\epsilon}\right)$$

Novel samples can be generated by iteratively applying the learned reverse transition from $t = T$ to $t = 0$.

### 3.4. Variance Schedule Augmentation

In Gaussian-Softmax diffusion, we observed that variance schedules cannot be used directly as-is due to the distortion introduced by the softmax projection on the injected noise. To address this, we propose an augmentation to the variance schedule, ensuring that the distribution of $\mathrm{argmax}(\mathbf{y}_t) \overset{.}{\sim} \mathcal{C}\left(\overline{\alpha_t}\mathbf{y}_0 + (1 - \overline{\alpha_t})/D\right)$. This is achieved through the following augmentation:

$$\overline{b_t} = \frac{f(\overline{\alpha_t})^2}{f(\overline{\alpha_t})^2 + f(k)^2}, \quad f(x) = \log\left(\frac{1 - x}{(D - 1)x + 1}\right) \quad (8)$$

Figure 2 illustrates the significance of this augmentation, showcasing its necessity over directly using the raw variance schedule. We also provide the derivation in Appendix A.4.

## 4. Sketch Diffusion

Since we focus solely on generating primitives, we represent a CAD sketch $X$ as a set of primitives $X = \{x_i\}$, where $X \in \mathbb{R}^{n \times d}$, $n$ denotes the maximum number of primitives, and $d$ represents the dimensionality of primitive features. Importantly, all $n!$ permutations of $X$ are equivalent in terms of geometry, since permuting $X$ doesn't change the actual geometry of each primitive. This invariance poses a challenge for generative modeling, as the ordering of primitives must not influence the learned generative process.

To address this, we adopt a permutation-equivariant diffusion methodology, ensuring that the denoising procedure remains consistent irrespective of primitive ordering. To

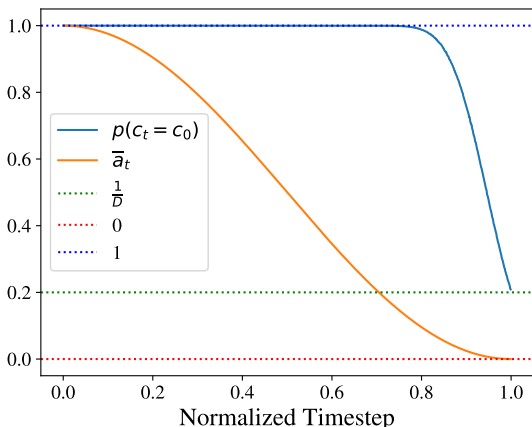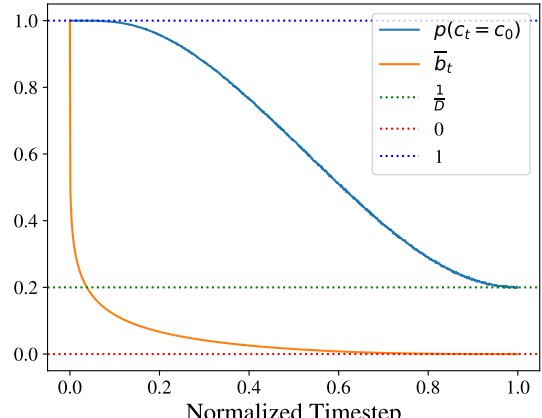

*Figure 2.* **Left:** The orange curve represents the raw cosine variance schedule $\overline{a_t}$, while the blue curve depicts the probability that the class label remains unchanged. **Right:** The orange curve shows the augmented variance schedule $\overline{b_t}$, as defined in Equation (8), and the blue curve again represents the probability of the class label not switching. The probabilities were calculated using Monte Carlo estimation over 100,000 samples. The augmented variance schedule results in a more gradual discrete forward process compared to the raw variance schedule.

apply noise to a sketch, we perturb each primitive independently, such that the forward transition distribution and cumulative transition distribution, respectively, factorize as:

$$p(X_{t+1} \mid X_t) = \prod_{i=1}^{n} p(x_{t+1,i} \mid x_{t,i}) \tag{9}$$

$$p(X_t \mid X_0) = \prod_{i=1}^{n} p(x_{t,i} \mid x_{0,i}) \tag{10}$$

Since each primitive undergoes the same noising process independently of the others, permuting the order of primitives in $X$, before the forward process, results in an equivalent permutation of the noised sketch $X_t$. That is, if $\pi$ is any permutation of $\{1, \ldots, n\}$, then $p(\pi(X_t) \mid \pi(X_0)) = p(X_t \mid X_0)$. Thus, the forward process itself is permutation-equivariant.

Applying Bayes' rule, the posterior distribution can be expressed as:

$$
\begin{aligned}
p(X_{t-1} \mid X_t, X_0) &= \frac{p(X_t|X_{t-1})p(X_{t-1}|X_0)}{p(X_t|X_0)} \\
&= \prod_{i=1}^{n} \frac{p(x_{t,i}|x_{t-1,i})p(x_{t-1,i}|x_{0,i})}{p(x_{t,i}|x_{0,i})} \\
&= \prod_{i=1}^{n} p(x_{t-1,i} \mid x_{t,i}, x_{0,i})
\end{aligned}
\tag{11}
$$

Thus, using the same line of reasoning as for the forward process, we can see that the reverse process is also permutation equivariant. To ensure the learned reverse process maintains permutation equivariance, the denoiser network must not introduce positional dependencies. This can be facilitated by using an architecture that is inherently permutation-equivariant. In our case, we adopt the diffusion transformer architecture by (Peebles & Xie, 2022) and simply omit positional encodings, since transformers have been shown to be intrinsically permutation-equivariant (Vaswani et al., 2017; Xu et al., 2024a). A similar architecture is used in Brepgen by (Xu et al., 2024b) for permutation invariant, instead of equivariant, generation of B-rep splines.

### 4.1. Primitive Diffusion

Each primitive is characterized by both discrete and continuous variables, necessitating a diffusion process that simultaneously operates in both domains. Formally, each primitive is represented as $x = (\mathbf{b}, \mathbf{c}, \mathbf{p})$, where $\mathbf{b}$ denotes the construction label, $\mathbf{c}$ the class label, and $\mathbf{p}$ the continuous parameters.

To model the diffusion process, we treat each component independently. The discrete variables $\mathbf{b}$ and $\mathbf{c}$ undergo a forward process governed by Equation 6 and a reverse process dictated by Equation 7. The continuous parameters $\mathbf{p}$, in contrast, follow a standard diffusion process, where the forward and reverse transitions are defined by Equations 2 and 3, respectively. Essentially, we perform joint discrete-continuous diffusion by independently performing Gaussian-Softmax diffusion for discrete variables and standard Gaussian diffusion for continuous parameters.

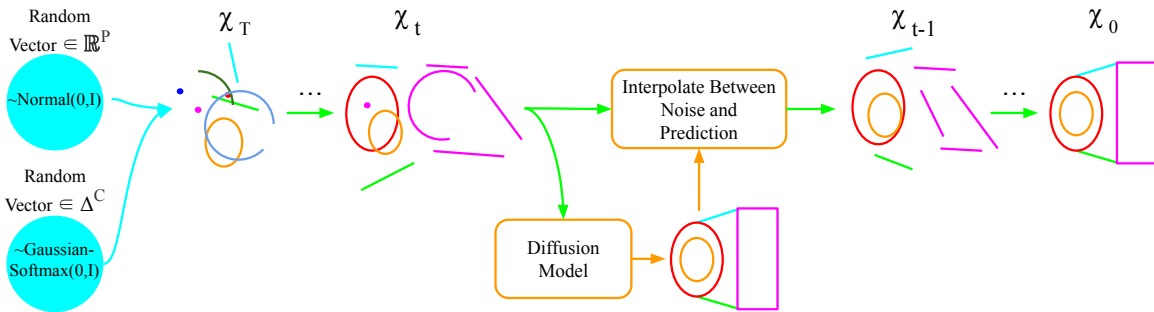

*Figure 3.* The generation pipeline of SketchDNN. Starting from a pure noise seed $\mathcal{X}_T$, the denoiser network iteratively refines the sample by interpolating between the noisy input and the model's prediction of the clean output. The final generated sketch, $\mathcal{X}_0$, is obtained after $T$ successive denoising steps. The interpolation formulae are given by Equation 7 for discrete variables and Equation 3 for continuous variables.

## 5. Experiments

We empirically validate our diffusion model, SketchDNN, against prior art. We also perform ablation studies to show that our approaches to address the heterogeneity and permutation invariance of sketches improves the performance of our model. We present metrics for likelihood and sample quality. Additionally we provide a qualitative comparison between our model and the dataset in Figure 4.

### 5.1. Dataset Preprocessing

We used the CAD sketch dataset introduced in SketchGraphs by Seff et al. (Seff et al., 2020), which consists of 15 million human-created CAD sketches extracted from Onshape, a cloud-based CAD platform. The dataset is highly imbalanced, with approximately 84% of sketches containing fewer than eight primitives. These sketches predominantly feature simple boxes that are almost identical to one another. To avoid biasing our model towards such sketches, we excluded sketches with fewer than eight primitives. Additionally, we discarded sketches with more than 16 primitives due to time and resource constraints. The sketches also exhibit significant variation in position and scale, ranging from a few millimeters to several meters in size. Accordingly, we applied a normalization procedure where each sketch was translated and rescaled, such that the bounding box was centered at the origin and inscribed within the unit square.

Next, the dataset contains a substantial number of duplicate sketches, primarily due to users following tutorials, reusing previously created designs, or leveraging Onshape's collaboration tools to replicate sketches. To again avoid bias, we grouped identical sketches and kept one sketch from each group. For this, we first performed 8-bit quantization on continuous parameters, then sorted the rows in the sketch matrix for a canonical ordering, and finally picked a corresponding unquantized sketch for each group. This was done to avoid sketches with identical geometry but differing orderings from being labeled as nonidentical. Lastly, we split the remaining 1.4 million CAD sketches into 3 subsets: we reserved 90% for training, 5% for validation, and 5% for testing. Finally, we simplified the parameterization of each primitive in accordance with Section 2. A similar procedure was performed by (Para et al., 2024) and (Seff et al., 2022).

### 5.2. Model and Training Setup

SketchDNN is based on a simplified variant of the diffusion transformer architecture proposed by Peebles et al. (Peebles & Xie, 2022), with positional encodings and gate conditioning omitted. The omission of positional encodings ensures that the denoiser network remains permutation equivariant, as discussed in Section 4. We use an embedding size of 512 with a depth of 32 transformer blocks. We trained our model to generate samples over $T = 2000$ timesteps. Since the diffusion process involves reconstructing a clean sketch from a noisy input, we use reconstruction loss, $\mathcal{L}_{\text{RECON}}$, composed of Mean-Squared Error (MSE) loss for continuous variables and Cross Entropy (CE) loss for discrete variables:

$$\mathcal{L}_{\text{RECON}} = \begin{cases} \lambda\mathcal{L}_{\text{MSE}} + \mathcal{L}_{\text{CE}} & \text{if } x \leq 150, \\ \mathcal{L}_{\text{MSE}} + \mathcal{L}_{\text{CE}} & \text{if } x > 150 \end{cases}$$

where we set $\lambda = 16$. We found weighing the MSE loss of smaller timesteps to be more beneficial to sample quality, than simply weighing the MSE loss higher over all timesteps. The model was trained for 1000 epochs using a batch size of $8 \times 512$, distributed across 8 NVIDIA A30 GPUs. A constant learning rate of $1 \times 10^{-4}$ was used throughout training.

To avoid hindering superposition, we use the ground truth primitive type to mask irrelevant parameters before computing the MSE loss. Thus, our denoiser network is trained to predict parameters for all possible primitive types and assign

confidence scores to each. For example, if the true primitive type is a line, the model still predicts parameters for other types, such as circles or arcs, but assigns probabilities reflecting its confidence in each prediction. As a result, during inference, we weight the continuous variables outputted by our model by their corresponding rescaled type probabilities. We rescale the predicted type probabilities by dividing each probability vector with its maximum element. This rescaling prevents relevant parameter values from decaying to zero throughout the reverse diffusion process while irrelevant parameters are masked out, ensuring more accurate predictions of noisy primitives. This adjustment is necessary since the model is trained to expect irrelevant parameters to approach zero at smaller timesteps, discrepancies may arise during the reverse process because we don't train the model to zero out irrelevant parameters. Applying this weighting ensures that the model's predictions better align with the ground truth, particularly at smaller timesteps.

## 6. Results

We evaluate SketchDNN against the most relevant prior methods for CAD sketch generation: SketchGen (Para et al., 2024) and Vitruvion (Seff et al., 2022). Both approaches represent CAD sketches as sequences of tokens and generate them autoregressively. Vitruvion, as the previous state of the art, serves as our primary baseline for comparison. Additionally, we assess the effectiveness of our diffusion-based approach by comparing it with conventional discrete diffusion methods as an ablation study, including categorical diffusion and latent diffusion. For latent diffusion, we train a variational autoencoder (VAE) to map sketches into a latent space and train an auxillary diffusion network to generate latents in a manner analogous to Stable Diffusion. For categorical diffusion, we train an alternative version of SketchDNN, which we term SketchDNN (Cat.), that is trained using the categorical diffusion paradigm by (Hoogeboom et al., 2021) while keeping all other architectural and training details identical. Similarly, we conduct an ablation study to examine the impact of permutation invariance in our generation process. To this end, we train an alternative version of SketchDNN, which we term SketchDNN (Pos.), that incorporates positional encodings while keeping all other aspects identical.

### 6.1. Negative Log-Likelihood

We evaluate the negative log-likelihood (NLL) as a measure of our model's ability to learn the distribution of CAD graphs. A lower NLL indicates better generalization and improved model performance; however, it does not necessarily correlate with sample quality. For diffusion models, computing the exact NLL is intractable, therefore, we follow the standard approach of approximating the NLL using the

*Table 1.* The NLL for each model is reported in bits, where a lower NLL is better. Notably, our diffusion model achieved SOA NLL. SketchDNN (Pos.) is the ablation that includes positional encodings, where SketchDNN (Cat.) is the ablation that uses categorical diffusion. Bold represents state of the art performance, while underlining represents second best performance.

| Method | Bits/Sketch↓ | Bits/Primitive↓ |
|---|---|---|
| SketchDNN (Ours) | **81.33** | **5.08** |
| SketchDNN (Pos.) | 83.03 | 5.18 |
| SketchDNN (Cat.) | 106.10 | 6.63 |
| Vitruvion | 84.80 | 8.19 |
| SketchGen | 88.22 | 8.60 |

ELBO, which satisfies the inequality $\text{ELBO} \geq \text{NLL}$. We calculate the NLL over our test set of 70K CAD sketches. The results are presented in Table 1. SketchDNN achieves a lower NLL than the previous state-of-the-art model, Vitruvion, demonstrating that our diffusion-based approach is better suited to CAD sketch generation than autoregressive modeling.

Notably, our Gaussian-Softmax diffusion paradigm significantly outperforms traditional categorical diffusion. In fact, SketchDNN (Cat.), which employs categorical diffusion, performs worse than both Vitruvion and Sketch-Gen. This suggests that the superposition mechanism in Gaussian-Softmax diffusion is a key factor in improving model performance. This finding is particularly surprising, as SketchDNN (Cat.), being a diffusion model, still has access to the full sketch context at any given time, yet performs worse. Additionally, our permutation-invariant diffusion paradigm appears to provide a measurable but smaller benefit, as indicated by the relatively close NLL scores of SketchDNN and SketchDNN (Pos.). These results suggest that while permutation-invariant denoising contributes to performance, the primary factor driving improvements in CAD sketch generation is the use of superposition.

*Table 2.* The FID, precision, and recall are presented for unconditional primitive generation. A lower FID is better, a higher precision is better, and a higher recall is better. We note that our diffusion model has achieved SOA FID and recall. SketchDNN (Pos.) is the ablation that includes positional encodings, where SketchDNN (Cat.) is the ablation that uses categorical diffusion. Bold represents state of the art performance, while underlining represents second best performance.

| Method | FID↓ | Precision↑ | Recall↑ |
|---|---|---|---|
| SketchDNN (Ours) | **7.80** | 0.246 | **0.266** |
| SketchDNN (Pos.) | 10.26 | 0.230 | 0.245 |
| Latent Diffusion | 93.34 | 0.134 | 0.033 |
| SketchDNN (Cat.) | 148.93 | 0.117 | 0.028 |
| Vitruvion | 16.04 | **0.294** | 0.176 |

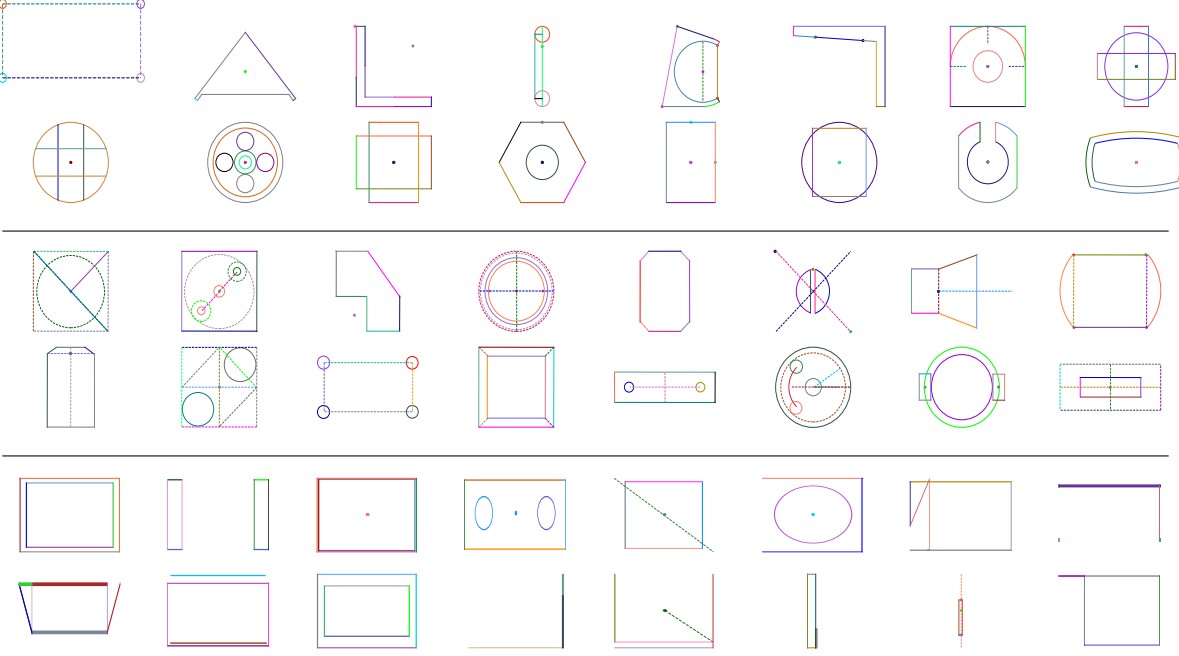

*Figure 4.* Comparison of CAD sketches from the SketchGraphs dataset (top), generations from SketchDNN (middle), and generations from Vitruvion (bottom).

## 6.2. Sample Quality

Here we present the Fréchet Inception Distance (FID), precision, and recall as metrics to evaluate the fidelity of generated CAD graphs, as shown in Table 2. Fidelity measures how closely synthetic CAD graphs resemble real CAD graphs. Standard image generation metrics are employed to assess sample quality, for this we render CAD sketches as monochromatic images. A lower FID score indicates higher fidelity samples, and our diffusion model achieves state-of-the-art (SOA) FID. Higher precision reflects a closer resemblance between generated and real samples, minimizing irrelevant or low-quality outputs, while higher recall indicates greater diversity in generated samples, capturing the full range of real data variations. Precision and recall often exhibit an inverse relationship, where improvements in one may lead to reductions in the other. As is standard in image generation literature, we compute these metrics using InceptionV3 over 10K CAD graphs from our test set.

We observe similar trends as in the NLL evaluation, with our Gaussian-Softmax diffusion paradigm again significantly outperforming alternative sketch generation approaches. Notably, SketchDNN (Cat.) yields the worst performance among all methods, further indicating that superposition plays a large role in CAD sketch generation. Surprisingly, the latent diffusion model also underperforms, indicating that latent space diffusion is ill suited for CAD sketch generation. Once again, SketchDNN and SketchDNN (Pos.)

achieve comparable results, reinforcing the conclusion that permutation-invariant denoising has a much smaller effect on generation quality than superposition. Since these models substantially outperform all other models, we conclude that the primary contributor to performance is the ability of our Gaussian-Softmax paradigm to accommodate superposition.

## 6.3. Failure Cases

Figure 5 illustrates three representative failure modes observed in SketchDNN. 1) In some cases, the model generates sketches lacking any discernible geometric structure or pattern. We find that this issue can be mitigated by increasing the weight of the model prediction prior to the reverse interpolation step in Equation 7, suggesting that high uncertainty in primitive type impairs the reverse process's ability to reconstruct coherent structure. We hypothesize that this failure mode could also be reduced through conditional generation, which would constrain the model's uncertainty during sampling. 2) A second failure involves primitives whose endpoints should be coincident but are instead separated by large gaps. This issue is likewise alleviated by increasing the contribution of the model prediction during the reverse step, indicating that the error stems from limitations in our parameter masking strategy. 3) The final failure mode involves the presence of redundant or extraneous primitives, often overlapping with valid geometry. We

traced this issue to our training data, where several sketches contain overlapping primitives.

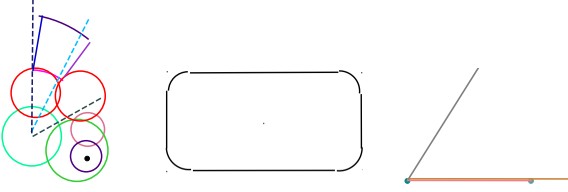

*Figure 5.* Visualizations of 3 primary failure cases. **Left:** Generated CAD sketch with no discernible pattern. **Middle:** Large gaps between primitives. **Right:** Extraneous primitives, zoom of the bottom left portion of a triangle where the bottom leg has an overlapping redundant line.

## 7. Related Work

Prior work in CAD sketch generation has predominantly focused on autoregressive approaches, treating sketches as sequences of tokens and leveraging language modeling techniques (Seff et al., 2020; Ganin et al., 2021; Para et al., 2024; Willis et al., 2021; Wu et al., 2021; Seff et al., 2022; Wu et al., 2025). In tangential domains with sketch-like data, for instance UI layout generation, DLT by (Levi et al., 2023) uses a joint continuous-discrete approach similar to ours except with Multinomial diffusion instead, whereas others solely use continuous diffusion (Shabani et al., 2022; Cheng et al., 2023). More broadly for CAD diffusion as a whole, researchers have primarily utilized latent diffusion to generate CAD models (Wang et al., 2024; Xu et al., 2022; 2024b).

Similarly, prior work on discrete diffusion has largely relied on the categorical distribution for discrete noise (Sohl-Dickstein et al., 2015; Hoogeboom et al., 2021; 2022; Lou et al., 2024; Bond-Taylor et al., 2021), or instead on applying continuous diffusion to discrete variables (Cohen et al., 2022; Yu et al., 2022; Han et al., 2022). Several researchers have proposed categorical relaxations outside the context of diffusion (Aitchison & Shen, 1980; Maddison et al., 2017; Jang et al., 2017; Potapczynski et al., 2020). (Maddison et al., 2017) and (Jang et al., 2017) propose a similar relaxation referred to as the Gumbel-Softmax distribution, where Gumbel, instead of Gaussian, vectors are mapped onto the probability simplex. (Potapczynski et al., 2020) use a pseudo-softmax transform to approximately map Gaussian values to the simplex, and (Aitchison & Shen, 1980) proposed a variant that expands the dimensionality of vectors by appending a zero before the softmax transform.

In the context of discrete diffusion, the closest works to our Gaussian-Softmax methodology are (Han et al., 2022) and (Karimi Mahabadi et al., 2024) which superficially resemble our proposed approach, however they differ in 3 significant

ways. 1) They use a heuristic reverse process whereas we follow a more principled approach and derive the reverse transition from the posterior of the forward process. 2) They use the cosine variance schedule directly which we demonstrate abruptly destroys information (see Figure 2). 3) As a result, they require a one-hot projection scheme that drastically hinders superposition which we show is important for CAD generation.

## 8. Conclusion

In this work, we present SketchDNN, the first data-space diffusion model for CAD sketch generation. We overcome the key challenges of sketch generation, namely the heterogeneity of primitive parameterizations and the permutation invariance of primitives, via superposition and permutation-invariant denoising, respectively. To facilitate this we propose Gaussian-Softmax diffusion, a novel discrete diffusion paradigm that enables blended class labels. In the same vein, we introduce the Gaussian-Softmax distribution a novel variant of the Logistic-Normal distribution. We demonstrate that our model advances the state-of-the-art in terms of both fidelity and diversity of sketches. We hope that future work will build off of framework to improve CAD sketch generation, or apply our framework to other domains such as text generation.

## Impact Statement

This paper presents work whose goal is to advance the field of Machine Learning. There are many potential societal consequences of our work, none which we feel must be specifically highlighted here.

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

## A. Gaussian-Softmax Distribution

The Gaussian-Softmax distribution ($\mathcal{GS}$), a variation of the Logistic-Normal distribution introduced by Aitchison (Aitchison & Shen, 1980), is the distribution of a Gaussian vector that has undergone the softmax transformation. The probability density is:

$$p(\vec{y}|\vec{\mu}, \sigma^2) = D^{-\frac{1}{2}} (2\pi\sigma^2)^{\frac{1-D}{2}} \left( \prod_{i=1}^{D} y_i \right)^{-1}$$

$$\times \exp \left\{ -\frac{1}{2\sigma^2} \left[ \sum_{i \neq D} \left( \log \frac{y_i}{y_D} - \mu_i + \mu_D \right)^2 - \frac{1}{D} \left( \sum_{i \neq D} \log \frac{y_i}{y_D} - \mu_i + \mu_D \right)^2 \right] \right\}$$

and the derivation of the density is provided in A.2.

### A.1. Derivation of Cumulative Forward Transition

A simple derivation for the cumulative transition is:

$$p\left( \vec{x}_{t+2} \mid \vec{x}_t \right) = \text{softmax} \left\{ \sqrt{\alpha_{t+2}} \log \left[ \text{softmax} \left\{ \sqrt{\alpha_{t+1}} \log (\vec{x}_t) + \sqrt{1 - \alpha_{t+1}} \vec{\epsilon} \right\} \right] + \sqrt{1 - \alpha_{t+2}} \vec{\epsilon} \right\}$$

Expand terms:

$$= \text{softmax} \left\{ \sqrt{\alpha_{t+2}} \left( \sqrt{\alpha_{t+1}} \log (\vec{x}_t) + \sqrt{1 - \alpha_{t+1}} \vec{\epsilon} + C \right) + \sqrt{1 - \alpha_{t+2}} \vec{\epsilon} \right\}$$

Reduce terms:

$$= \text{softmax} \left\{ \sqrt{\alpha_{t+2}\alpha_{t+1}} \log (\vec{x}_t) + \sqrt{\alpha_{t+2}(1 - \alpha_{t+1})} \vec{\epsilon} + \sqrt{\alpha_{t+2}} C + \sqrt{1 - \alpha_{t+2}} \vec{\epsilon} \right\}$$

Constants disappear due to shift invariance of softmax:

$$= \text{softmax} \left\{ \sqrt{\alpha_{t+2}\alpha_{t+1}} \log (\vec{x}_t) + \sqrt{\alpha_{t+2}(1 - \alpha_{t+1})} \vec{\epsilon} + \sqrt{1 - \alpha_{t+2}} \vec{\epsilon} \right\}$$

Sum of Gaussians is another Gaussian with summed variances:

$$= \text{softmax} \left\{ \sqrt{\alpha_{t+2}\alpha_{t+1}} \log (\vec{x}_t) + \sqrt{\alpha_{t+2}(1 - \alpha_{t+1}) + 1 - \alpha_{t+2}} \vec{\epsilon} \right\}$$

Merging terms accumulates the $a$ terms:

$$= \text{softmax} \left\{ \sqrt{\alpha_{t+2}\alpha_{t+1}} \log (\vec{x}_t) + \sqrt{1 - \alpha_{t+2}\alpha_{t+1}} \vec{\epsilon} \right\}$$

therefore iteratively applying the forward transition will simply accumulate the variance schedule terms.

### A.2. Derivation of Gaussian Softmax Density

Our strategy is to use the change of variables formula:

$$p'(\vec{y}) = p\left( h^{-1}(\vec{y}) \right) \text{Det} \left[ J\left( \vec{h}(\vec{y}) \right) \right]$$

where $\vec{h}(\vec{y})$ is some invertible function and $\text{Det}(J(\vec{h}(\vec{y}))$ is the determinant of the jacobian. More specifically,

$$\text{softmax}\{\vec{y}\} = \text{softmax}\{\vec{y} - \vec{1} \cdot y_D\}$$

holds due to the shift invariance of softmax, thus our strategy is to first find the density of $\vec{y'} = [y_1 - y_D, y_2 - y_D, ..., 0]$, as this "centered" form turns the softmax into an invertible function $\vec{h}(\vec{y'}) = \text{softmax}\{\vec{y'}\}$ where the inverse is $\vec{h}^{-1}(\vec{x}) = \log(\vec{x}/x_D)$. The derivation is as follows for $\vec{y} \sim \mathcal{N}(\vec{\mu}, \vec{\sigma}^2 \mathbf{I})$, additionally for brevity we aggregate all factors into a single variable $C$:

Marginalizing over $y_D$ yields the density of the "centered" density:

$$p(\vec{y'}) = \int_{-\infty}^{\infty} p(\vec{y'} \mid y_D)p(y_D) \, dy_D$$

$$= \int_{-\infty}^{\infty} \left[ \prod_{i \neq D} (2\pi\sigma^2)^{-\frac{1}{2}} \exp\left\{ -\frac{1}{2\sigma^2}(y_i - y_D - \mu_i)^2 \right\} \right] \left[ (2\pi\sigma^2)^{-\frac{1}{2}} \exp\left\{ -\frac{1}{2\sigma^2}(y_D - \mu_D)^2 \right\} \right] dy_D$$

Expand terms:

$$= (2\pi\sigma^2)^{-\frac{D}{2}} \int_{-\infty}^{\infty} \left[ \prod_{i \neq D} \exp\left\{ -\frac{1}{2\sigma^2}(y_i^2 - 2y_i\mu_i + \mu_i^2) \right\} \right]$$

$$\times \exp\left\{ -\frac{1}{2\sigma^2} \left( y_D^2 - 2y_D\mu_D + \mu_D^2 + (D-1)y_D^2 + 2y_D \sum_{i \neq D} \mu_i - y_i \right) \right\} dy_D$$

Reduce $y_D$ terms:

$$= C \left[ \prod_{i \neq D} \exp\left\{ -\frac{1}{2\sigma^2}(y_i - \mu_i)^2 \right\} \right]$$

$$\times \int_{-\infty}^{\infty} \exp\left\{ -\frac{1}{2\sigma^2} \left( y_D^2 - 2y_D\mu_D + \mu_D^2 + (D-1)y_D^2 + 2y_D \sum_{i \neq D} \mu_i - y_i \right) \right\} dy_D$$

Reduce terms:

$$= C \int_{-\infty}^{\infty} \exp\left\{ -\frac{1}{2\sigma^2} \left( Dy_D^2 - 2y_D \left( \mu_D + \sum_{i \neq D} y_i - \mu_i \right) + \mu_D^2 \right) \right\} dy_D$$

Complete the square:

$$= C \exp\left\{ -\frac{1}{2\sigma^2}\mu_D^2 \right\} \int_{-\infty}^{\infty} \exp\left\{ -\frac{1}{2\sigma^2} \left( Dy_D^2 - 2y_D \left( \mu_D + \sum_{i \neq D} y_i - \mu_i \right) \right) \right\} dy_D$$

$$= C \int_{-\infty}^{\infty} \exp\left\{ -\frac{1}{2\sigma^2} \left( \sqrt{D}y_D - \frac{\left( \mu_D + \sum_{i \neq D} y_i - \mu_i \right)}{\sqrt{D}} \right)^2 \right\} dy_D$$

The integrand is a Gaussian density in terms of $y_D$:

$$= (2\pi\sigma^2)^{-\frac{D}{2}} \left[ \prod_{i \neq D} \exp\left\{ -\frac{1}{2\sigma^2}(y_i - \mu_i)^2 \right\} \right] \exp\left\{ -\frac{1}{2\sigma^2}\mu_D^2 \right\}$$

$$\times \exp\left\{ -\frac{1}{2\sigma^2} \frac{\left( \mu_D + \sum_{i \neq D} y_i - \mu_i \right)^2}{D} \right\} \sqrt{\frac{2\pi\sigma^2}{D}}$$

We can simplify this further using the fact that shifting the mean by any constant scalar does not affect the density due to the shift invariance of the softmax operation i.e., $p(\vec{y}|\vec{u}, \sigma^2) = p(\vec{y}|\vec{u} + c\vec{1}, \sigma^2)$, thus if we use $c = -\mu_D$ the density becomes:

$$= D^{-\frac{1}{2}} \left(2\pi\sigma^2\right)^{\frac{1-D}{2}} \exp\left\{ -\frac{1}{2\sigma^2} \left[ \sum_{i \neq D} (y_i - \mu_i + \mu_D)^2 - \frac{1}{D} \left( \sum_{i \neq D} y_i - \mu_i + \mu_D \right)^2 \right] \right\}$$

Now that we have the density of $\vec{y'}$ we can use a straightforward application of the change of variables formula, with the known result that the determinant of the Jacobian of the softmax is $(\prod_i^D y_i)^{-1}$ (Jang et al., 2017; Maddison et al., 2017) to obtain:

$$p(\vec{y}|\vec{\mu}, \sigma^2) = D^{-\frac{1}{2}} (2\pi\sigma^2)^{\frac{1-D}{2}} \left( \prod_{i=1}^D y_i \right)^{-1}$$

$$\times \exp\left\{ -\frac{1}{2\sigma^2} \left[ \sum_{i \neq D} \left( \log \frac{y_i}{y_D} - \mu_i + \mu_D \right)^2 - \frac{1}{D} \left( \sum_{i \neq D} \log \frac{y_i}{y_D} - \mu_i + \mu_D \right)^2 \right] \right\}$$

### A.3. Derivation of Posterior for Guassian-Softmax

For the reverse process we sample from the posterior distribution $p(\vec{x}_{t-1}|\vec{x}_t, \vec{x}_0)$, using the same setup as in DDPM (Ho et al., 2020):

$$p(\vec{x}_{t-1}|\vec{x}_t, \vec{x}_0) = \frac{p(\vec{x}_{t-1}, \vec{x}_t, \vec{x}_0)}{p(\vec{x}_t, \vec{x}_0)} = \frac{p(\vec{x}_t|\vec{x}_{t-1}, \vec{x}_0)p(\vec{x}_{t-1}, \vec{x}_0)}{p(\vec{x}_t, \vec{x}_0)} = \frac{p(\vec{x}_t|\vec{x}_{t-1}, \vec{x}_0)p(\vec{x}_{t-1}|\vec{x}_0)}{p(\vec{x}_t|\vec{x}_0)}$$

and due to the Markov property $p(\vec{x}_t|\vec{x}_{t-1}, \vec{x}_0) = p(\vec{x}_t|\vec{x}_{t-1})$ the posterior simplifies to:

$$p(\vec{x}_{t-1}|\vec{x}_t, \vec{x}_0) = \frac{p(\vec{x}_t|\vec{x}_{t-1})p(\vec{x}_{t-1}|\vec{x}_0)}{p(\vec{x}_t|\vec{x}_0)} \propto p(\vec{x}_t|\vec{x}_{t-1})p(\vec{x}_{t-1}|\vec{x}_0)$$

fortunately we have access to the necessary densities which are simply:

1.

$$p(\vec{x}_{t-1}|\vec{x}_0) \propto \left( \prod_i^D x_{t-1,i} \right)^{-1} \exp\left[ -\frac{1}{2(1 - \bar{a}_{t-1})} \sum_{i \neq D} v_i^2 \right] \exp\left[ \frac{1}{2D(1 - \bar{a}_{t-1})} \left( \sum_{i \neq D} v_i \right)^2 \right]$$

where $v_i = \log \frac{x_{t-1,i}}{x_{t-1,D}} - \sqrt{\bar{a}_{t-1}} \log \frac{x_{0,i}}{x_{0,D}}$

2.

$$p(\vec{x}_t|\vec{x}_{t-1}) \propto \exp\left[ -\frac{1}{2(1 - a_t)} \sum_{i \neq D} r_i^2 \right] \exp\left[ \frac{1}{2D(1 - a_t)} \left( \sum_{i \neq D} r_i \right)^2 \right]$$

where $r_i = \log \frac{x_{t,i}}{x_{t,D}} - \sqrt{a_t} \log \frac{x_{t-1,i}}{x_{t-1,D}}$

Focusing on the first exponential terms with simplified notation where $z_i = \log \frac{x_{t,i}}{x_{t,D}}$, $y_i = \log \frac{x_{t-1,i}}{x_{t-1,D}}$, $x_i = \log \frac{x_{0,i}}{x_{0,D}}$, $\sigma_t^2 = 1 - a_t$, and $\overline{\sigma}_{t-1}^2 = 1 - \overline{a}_{t-1}$

$$\exp\left[-\frac{1}{2(1-\overline{a}_{t-1})}\sum_{i\neq D} v_i^2\right] \exp\left[-\frac{1}{2(1-a_t)}\sum_{i\neq D} r_i^2\right]$$

$$= \exp\left\{-\frac{1}{2\overline{\sigma}_{t-1}^2}\sum_{i\neq D}\left(y_i - \sqrt{\overline{a}_{t-1}}x_i\right)^2 - \frac{1}{2\sigma_t^2}\sum_{i\neq D}\left(z_i - \sqrt{a_t}y_i\right)^2\right\}$$

$$= \exp\left\{-\frac{1}{2}\left[\sum_{i\neq D}\frac{1}{\overline{\sigma}_{t-1}^2}\left(y_i - \sqrt{\overline{a}_{t-1}}x_i\right)^2 + \frac{1}{\sigma_t^2}\left(z_i - \sqrt{a_t}y_i\right)^2\right]\right\}$$

$$= \exp\left\{-\frac{1}{2}\left[\sum_{i\neq D}\frac{1}{\overline{\sigma}_{t-1}^2}\left(y_i^2 - 2y_i\sqrt{\overline{a}_{t-1}}x_i + \overline{a}_{t-1}x_i^2\right)\right.\right.$$
$$\left.\left. + \frac{1}{\sigma_t^2}\left(z_i^2 - 2\sqrt{a_t}y_i z_i + a_t y_i^2\right)\right]\right\}$$

$$\propto \exp\left\{-\frac{1}{2}\left[\sum_{i\neq D}\frac{y_i^2}{\overline{\sigma}_{t-1}^2} - \frac{2y_i\sqrt{\overline{a}_{t-1}}x_i}{\overline{\sigma}_{t-1}^2} - \frac{2\sqrt{a_t}y_i z_i}{\sigma_t^2} + \frac{a_t y_i^2}{\sigma_t^2}\right]\right\}$$

$$= \exp\left\{-\frac{1}{2}\left[\sum_{i\neq D}\left(\frac{1}{\overline{\sigma}_{t-1}^2} + \frac{a_t}{\sigma_t^2}\right)y_i^2 - 2\left(\frac{\sqrt{\overline{a}_{t-1}}x_i}{\overline{\sigma}_{t-1}^2} + \frac{\sqrt{a_t}z_i}{\sigma_t^2}\right)y_i\right]\right\}$$

observe that we can read the posterior mean and variance as:

$$\mu_{t-1,i} = \left(\frac{\sqrt{\overline{a}_{t-1}}x_i}{\overline{\sigma}_{t-1}^2} + \frac{\sqrt{a_t}z_i}{\sigma_t^2}\right)\sigma_{t-1}^2 = \frac{\sqrt{a_t}(1-\overline{a}_{t-1})z_i + \sqrt{\overline{a}_{t-1}}(1-a_t)x_i}{1-\overline{a}_t}$$

$$\sigma_{t-1}^2 = \left(\frac{1}{\overline{\sigma}_{t-1}^2} + \frac{a_t}{\sigma_t^2}\right)^{-1} = \frac{(1-a_t)(1-\overline{a}_{t-1})}{1-\overline{a}_t}$$

since the form has to be proportional to $\exp\left\{-\frac{1}{2\sigma^2}(y_i - \mu)^2\right\}$.

Similarly for the second exponential term:

$$\exp\left[\frac{1}{2D(1-\overline{a}_{t-1})}\left(\sum_{i\neq D} v_i\right)^2\right]\exp\left[\frac{1}{2D(1-a_t)}\left(\sum_{i\neq D} r_i\right)^2\right]$$

$$= \exp\left\{\frac{1}{2D}\left[\frac{1}{\overline{\sigma}_{t-1}^2}\left(\sum_{i\neq D} y_i - \sqrt{\overline{a}_{t-1}}x_i\right)^2 + \frac{1}{\sigma_t^2}\left(\sum_{i\neq D} z_i - \sqrt{a_t}y_i\right)^2\right]\right\}$$

$$= \exp\left\{\frac{1}{2D}\left[\frac{1}{\overline{\sigma}_{t-1}^2}\left(\sum_{\mathbf{i\neq D}}(\mathbf{y_i} - \sqrt{\mathbf{\overline{a}_{t-1}}}\mathbf{x_i})^2 + 2\sum_{j<i,i\neq D}(y_i - \sqrt{\overline{a}_{t-1}}x_i)(y_j - \sqrt{\overline{a}_{t-1}}x_j)\right)\right.\right.$$
$$\left.\left. + \frac{1}{\sigma_t^2}\left(\sum_{\mathbf{i\neq D}}(\mathbf{z_i} - \sqrt{\mathbf{a_t}}\mathbf{y_i})^2 + 2\sum_{j<i,i\neq D}(z_i - \sqrt{a_t}y_i)(z_j - \sqrt{a_t}y_j)\right)\right]\right\}$$

The terms in bold correspond exactly with the first exponential term, and imply the same posterior mean and variance,

further more the remaining terms are proportional to:

$$\exp \frac{1}{2D} \left[ 2 \sum_{j<i,i\neq D} \frac{1}{\overline{\sigma}_{t-1}^2} \left( y_i y_j - \sqrt{\overline{a}_{t-1}} x_j y_i - \sqrt{\overline{a}_{t-1}} x_i y_j \right) + \frac{1}{\sigma_t^2} \left( -\sqrt{a_t} z_i y_j - \sqrt{a_t} z_j y_i + a_t y_i y_j \right) \right]$$

$$= \exp \frac{1}{2D} \left[ 2 \sum_{j<i,i\neq D} \frac{1}{\overline{\sigma}_{t-1}^2} \left( y_i y_j - \sqrt{\overline{a}_{t-1}} x_j y_i - \sqrt{\overline{a}_{t-1}} x_i y_j \right) + \frac{1}{\sigma_t^2} \left( -\sqrt{a_t} z_i y_j - \sqrt{a_t} z_j y_i + a_t y_i y_j \right) \right]$$

$$= \exp \frac{1}{2D} \left[ 2 \sum_{j<i,i\neq D} \left( \frac{1}{\overline{\sigma}_{t-1}^2} + \frac{a_t}{\sigma_t^2} \right) y_i y_j - \left( \frac{\sqrt{\overline{a}_{t-1}} x_j}{\overline{\sigma}_{t-1}^2} + \frac{\sqrt{a_t} z_j}{\sigma_t^2} \right) y_i - \left( \frac{\sqrt{\overline{a}_{t-1}} x_i}{\overline{\sigma}_{t-1}^2} + \frac{\sqrt{a_t} z_i}{\sigma_t^2} \right) y_j \right]$$

which again imply the same posterior mean and variance since the form has to be proportional to:

$$\exp \left\{ 2 \left[ \sum_{j<i,i\neq D} \frac{y_i y_j}{\sigma^2} - \frac{\mu_j}{\sigma^2} y_i - \frac{\mu_i}{\sigma^2} y_j + \frac{\mu_i \mu_j}{\sigma^2} \right] \right\}$$

Thus all terms agree on the same posterior mean and variance of:

$$\mu_{t-1,i} = \left( \frac{\sqrt{\overline{a}_{t-1}} x_i}{\overline{\sigma}_{t-1}^2} + \frac{\sqrt{a_t} z_i}{\sigma_t^2} \right) \sigma_{t-1}^2 = \frac{\sqrt{a_t}(1-\overline{a}_{t-1})z_i + \sqrt{\overline{a}_{t-1}}(1-a_t)x_i}{1-\overline{a}_t}$$

$$\sigma_{t-1}^2 = \left( \frac{1}{\overline{\sigma}_{t-1}^2} + \frac{a_t}{\sigma_t^2} \right)^{-1} = \frac{(1-a_t)(1-\overline{a}_{t-1})}{1-\overline{a}_t}$$

We can simplify the posterior mean by utilizing the shift invariance property, where we observe that:

$$\mu_{t-1,i} = \frac{\sqrt{a_t}(1-\overline{a}_{t-1})(\log x_{t,i} - \log x_{t,D}) + \sqrt{\overline{a}_{t-1}}(1-a_t)(\log x_{0,i} - \log x_{0,D})}{1-\overline{a}_t}$$

$$= \frac{\sqrt{a_t}(1-\overline{a}_{t-1})\log x_{t,i} + \sqrt{\overline{a}_{t-1}}(1-a_t)\log x_{0,i}}{1-\overline{a}_t} - \frac{\sqrt{a_t}(1-\overline{a}_{t-1})\log x_{t,D} + \sqrt{\overline{a}_{t-1}}(1-a_t)\log x_{0,D}}{1-\overline{a}_t}$$

$$= \frac{\sqrt{a_t}(1-\overline{a}_{t-1})\log x_{t,i} + \sqrt{\overline{a}_{t-1}}(1-a_t)\log x_{0,i}}{1-\overline{a}_t} + C$$

so the posterior mean can be simplified as:

$$\mu_{t-1,i} = \frac{\sqrt{a_t}(1-\overline{a}_{t-1})\log x_{t,i} + \sqrt{\overline{a}_{t-1}}(1-a_t)\log x_{0,i}}{1-\overline{a}_t}$$

Since all the terms agree on the same posterior mean and variance, and furthermore the posterior density has the same form as the Gaussian-Softmax distribution, we can conclude that $p(\vec{x}_{t-1}|\vec{x}_t, \vec{x}_0) = p(\vec{x}_{t-1}|\vec{\mu}_{t-1}, \sigma_{t-1}^2 \mathbf{I})$

### A.4. Derivation of Variance Schedule Augmentation

As shown in Figure 2, we need to augment our chosen variance schedule to ensure that the class labels are gradually noised. Taking inspiration from Categorical diffusion, our desideratum is to smoothly noise the class label such that the argmax of $\mathbf{x}_t$ follows the distribution $\mathcal{C}(\overline{b}_t \mathbf{x}_0 + \frac{1}{D}(1-\overline{b}_t)\mathbf{1})$ where $\mathcal{C}$ is the categorical distribution and $b_0, b_1, \ldots, b_T$ is a variance schedule of our choosing. Unfortunately there is no closed form formula to determine the argmax of a Gaussian vector, so we instead approximate a Gaussian vector with a Gumbel vector. A useful property of the Gumbel distribution is that it can be used to reparameterize the Categorical distribution where $\text{argmax}\{a \log \mathbf{p} + \mathbf{g}\} \sim \mathcal{C}(\text{softmax}\{a \log \mathbf{p}\})$, $\mathbf{g} \sim \mathcal{G}(0,1)$, and $\mathcal{G}$ is the Gumbel distribution (Huijben et al., 2021). Then considering the forward process in Gaussian-Softmax diffusion we can derive:

$$\text{argmax}\{\mathbf{x}_t\} \approx \text{argmax} \left\{ \text{softmax} \left( \sqrt{\overline{\alpha_t}} \log \left( k\mathbf{x}_0 + \frac{1-k}{D} \right) + \sqrt{1-\overline{\alpha_t}}\mathbf{g} \right) \right\}.$$

Simplifying the expression inside the argmax:

$$= \text{argmax} \left\{ \sqrt{\overline{\alpha_t}} \log \left( k\mathbf{x}_0 + \frac{1-k}{D}\mathbf{1} \right) + \sqrt{1-\overline{\alpha_t}}\mathbf{g} \right\}$$

$$= \text{argmax} \left\{ \sqrt{\frac{\overline{\alpha_t}}{1-\overline{\alpha_t}}} \log \left( k\mathbf{x}_0 + \frac{1-k}{D}\mathbf{1} \right) + \mathbf{g} \right\} \sim \text{softmax} \left\{ \sqrt{\frac{\overline{\alpha_t}}{1-\overline{\alpha_t}}} \log \left( k\mathbf{x}_0 + \frac{1-k}{D}\mathbf{1} \right) \right\}$$

We aim to satisfy:

$$\text{softmax} \left\{ \sqrt{\frac{\overline{\alpha_t}}{1-\overline{\alpha_t}}} \log \left( k\mathbf{x}_0 + \frac{1-k}{D}\mathbf{1} \right) \right\} = \bar{b}_t \mathbf{x}_0 + \frac{1-\bar{b}_t}{D}\mathbf{1}$$

Taking the logarithm of both sides gives:

$$\sqrt{\frac{\overline{\alpha_t}}{1-\overline{\alpha_t}}} \log \left( k\mathbf{x}_0 + \frac{1-k}{D}\mathbf{1} \right) + \mathbf{c} = \log \left( \bar{b}_t \mathbf{x}_0 + \frac{1-\bar{b}_t}{D}\mathbf{1} \right)$$

Assuming without loss of generality that $\mathbf{x}_0 = [1, 0, \ldots, 0]$, we have:

$$\sqrt{\frac{\overline{\alpha_t}}{1-\overline{\alpha_t}}} \left[ \log \left( k + \frac{1-k}{D} \right), \log \left( \frac{1-k}{D} \right), \ldots \right] + \mathbf{c} = \left[ \log \left( \bar{b}_t + \frac{1-\bar{b}_t}{D} \right), \log \left( \frac{1-\bar{b}_t}{D} \right), \ldots \right]$$

Since $\mathbf{c}$ is a free parameter, we can set:

$$\mathbf{c} = \left[ \log \left( \bar{b}_t + \frac{1-\bar{b}_t}{D} \right) - \sqrt{\frac{\overline{\alpha_t}}{1-\overline{\alpha_t}}} \log \left( k + \frac{1-k}{D} \right) \right] \mathbf{1}$$

This reduces the equation to:

$$\sqrt{\frac{\overline{\alpha_t}}{1-\overline{\alpha_t}}} \left[ 0, \log \left( \frac{1-k}{(D-1)k+1} \right), \ldots \right] = \left[ 0, \log \left( \frac{1-\bar{b}_t}{(D-1)\bar{b}_t+1} \right), \ldots \right]$$

Thus, we deduce:

$$\sqrt{\frac{\overline{\alpha_t}}{1-\overline{\alpha_t}}} = \log \left( \frac{1-\bar{b}_t}{(D-1)\bar{b}_t+1} \right) \Big/ \log \left( \frac{1-k}{(D-1)k+1} \right)$$

Finally, isolating $\overline{\alpha_t}$ yields:

$$\overline{\alpha_t} = \frac{n^2}{n^2 + m^2}, \text{ where } n = \log \left( \frac{1-\bar{b}_t}{(D-1)\bar{b}_t+1} \right), \quad m = \log \left( \frac{1-k}{(D-1)k+1} \right).$$

