# OpenReview forum: "SketchDNN: Joint Continuous-Discrete Diffusion for CAD Sketch Generation"
_ICML.cc/2025/Conference — ICML 2025 poster_

### Official Review · Reviewer_c8Zn · 2025-03-08

**Overall Recommendation:** 3

**Summary:**

The paper proposes a diffusion based CAD sketch generation framework using a mixture of continuous and discrete diffusion. Technically, it introduces Gaussian-softmax diffusion, which is able to model categorical distributions with diffusion models. The paper provides a detailed theoretical derivation for their proposed Gaussian softmax diffusion and shows its effectiveness in generating CAD sketches. Experimentally, the paper shows state-of-the-art CAD generation results compared to baselines.

## Post Rebuttal
I appreciate the authors' rebuttal addressing many of my concerns. I'd like to keep my rating as it is. The main concern I have is still the usability of the method given that it doesn't support any kinds of controllable generation. Thus, a discussion on this front would be great to be included in the final revised edition.

**Claims And Evidence:**

The paper proposes a novel diffusion process with Gaussian-softmax diffusion. Using it, the paper is able to generate state-of-the-art CAD sketches that contain both continuous and discrete parameters. While this claim is supported by the paper's superior CAD sketch generation results compared to baselines quantitatively, it would be great to also see qualitative examples compared with the baselines.

Further, the paper also claims that its Gaussian-softmax diffusion works better than categorical diffusion. An ablation study comparing the two would be good to have.

**Essential References Not Discussed:**

N/A

**Experimental Designs Or Analyses:**

The paper uses a standard dataset following the existing data pre-processing scheme. The paper only presents unconditional generation results, and lack qualitative comparisons with existing baselines. Further, it's unclear how can we use the model for downstream tasks, since it lacks any type of control

**Methods And Evaluation Criteria:**

The paper's usage of Gaussian-softmax diffusion is convincing for the task of CAD sketch generation. The paper also uses an existing dataset with standard generation metrics. I would appreciate a more qualitative comparison with existing methods.

While it's standard to limit the number of primitives to below 16, I'm curious to see the scalability of this method when the number of primitives increases.

Furthermore, I'm a little puzzled by the independence assumption in the reverse process of the diffusion. The decomposition in Eq.13-14 assumes the independence of each primitive during the forward and reverse processes. Then does that mean the denoising process of each primitive is independent from each other? Then, it would become a primitive generation rather than a sketch generation.

**Other Comments Or Suggestions:**

Ln.113 departure -> depart.

**Other Strengths And Weaknesses:**

THe paper is techically sound with validations backing up the theoretical innovations. I don't have any additional weaknesses besides things I listed above.

**Questions For Authors:**

I have listed my questions in the previous sections. Specifically, I would like to see
1. More qualitative results compared with baselines.
2. Clarification on the independence assumption among primitives during the denoising process.
3. Discussion on the scalability of the method when increasing the number of primitives modeled.

**Relation To Broader Scientific Literature:**

The paper provides a novel diffusion formulation using Gaussian Softmax distributions. It's a good extension to the current diffusion model literation in terms of modeling categorical distributions.

**Theoretical Claims:**

The paper provides an extensive set of derivations for their Gaussian-softmax diffusion process. The math looks convincing, but I didn't look into the details too carefully.

---

> ### Author Rebuttal · Authors · 2025-03-31
>
> 1. While this claim is supported by the paper's superior CAD sketch generation results compared to baselines quantitatively, it would be great to also see qualitative examples compared with the baselines.
>    1. We have a qualitative analysis written and ready for the final paper. However, we only compare ours against Vitruvion, as other prior arts like SketchGen don’t have published code. We chose not to do a qualitative analysis against Sketchgraphs since Vitruvion is a follow up work to it.
> 2. The paper also claims that its Gaussian-softmax diffusion works better than categorical diffusion. An ablation study comparing the two would be good to have.
>    1. We do an ablation study see SketchDNN (Cat.) in Table 1 and Table 2\. Which is our SketchDNN model but just using methodology set by Hoogeboom et al. in “Argmax Flows and Multinomial Diffusion: Learning Categorical Distributions” which doesn’t allow for superposition.
> 3. While it's standard to limit the number of primitives to below 16, I'm curious to see the scalability of this method when the number of primitives increases.
>    1. Since the backbone of our model is essentially just the DiT model by Peebles & Xie, it has the same scaling laws as that of DiT.
> 4. Furthermore, I'm a little puzzled by the independence assumption in the reverse process of the diffusion. The decomposition in Eq.13-14 assumes the independence of each primitive during the forward and reverse processes. Then does that mean the denoising process of each primitive is independent from each other? Then, it would become a primitive generation rather than a sketch generation.
>    1. The primitives can only be independently noised and denoised when given the ground truth or model prediction of the noiseless CAD sketch. The noiseless CAD sketch is just like the noise prediction in image diffusion models, which also denoise each pixel independently.
> 5. Further, it's unclear how we can use the model for downstream tasks, since it lacks any type of control.
>    1. You are correct, we do not show conditional generation. Perhaps we used the wrong terminology, this will be fixed in the refined paper. What we meant was that the primitives generated by our model can be outfitted with constraints using an auto constrain model and then be parsed into Onshape.

---

### Official Review · Reviewer_5qBS · 2025-03-19

**Overall Recommendation:** 2

**Summary:**

Authors proposed a diffusion-based generative model for 2D CAD engineering drawings of parametric primitives. Main contributions include the use of Gaussian-softmax to unify discrete and continuous parameters, which works well for a modified diffusion model without positional encoding. And also a superposition data representation that joins the parameters from different types of parameters all in one. Results are reported for SketchGraphs where the generation quality are better compared to baselines.

**Claims And Evidence:**

Line 108: “we propose the first diffusion-based generative model for parametric CAD sketches” --- This is inaccurate, there is already existing works that apply diffusion model to generate CAD sketches. E.g “Diffusion-CAD: Controllable Diffusion Model for Generating Computer-Aided Design Models”, they generate both the CAD sketches and also the extrusion parameters that turn it into a CAD model. Similarly, there is also "VQ-CAD: Computer-Aided Design model generation with vector quantized diffusion".

**Essential References Not Discussed:**

Related work section is too short and does not cover previous works.  Missing citations for sketch-and-extrude CAD generation:

1) Diffusion-CAD: Controllable Diffusion Model for Generating Computer-Aided Design Models

2) VQ-CAD: Computer-Aided Design model generation with vector quantized diffusion

3) SkexGen: Autoregressive Generation of CAD Construction Sequences with Disentangled Codebooks


Also missing citations of discrete and continuous diffusion for CAD-related or vector-like data:

4) HouseDiffusion: Vector Floorplan Generation via a Diffusion Model with Discrete and Continuous Denoising

5) CoLay: Controllable Layout Generation through Multi-conditional Latent Diffusion

6) PLay: Parametrically Conditioned Layout Generation using Latent Diffusion


Finally removing the positional encoding in CAD generation has already be done in BrepGen and shown to address the permutation invariant nature of CAD models (section 6.3). This should also be properly acknowledged.

7) BrepGen: A B-rep Generative Diffusion Model with Structured Latent Geometry

**Experimental Designs Or Analyses:**

Baseline resutls from Vitruvion (table 1) is not the same as those reported in the original paper. I believe setting is the same, 16-max primitives. But fig 5 in Vitruvion reports better performance of 6.35 (per primitives) and 66.6 (per sketch).


Analysis for the benfit of superposition (line 339-344) also lacks proper support. SketchDNN (Cat) used discrete diffusion and in theory should be much worse than continous diffusion. I am not surprised about the results. However this is not a valid proof that superposition mechanism is a key factor for performance. For that, I am expecting to see a comparision where different primitives are seperated into different tokens, with only parameters related to that primitive type included.

**Methods And Evaluation Criteria:**

Paper has no novel and uniqueness scores, which is common in CAD generation. Also no closest-retrieval results to demonstrate model is not overfitting to training set (something like fig 6 in Neural Wavelet-domain Diffusion for 3D Shape Generation will be nice to have).

**Other Comments Or Suggestions:**

Minor type Line 380 state-of-the-art (SOA)  -> SOTA

**Other Strengths And Weaknesses:**

Data representation is similar to DeepCAD and Vitruvion without too many modifications, but the Gaussian-Softmax that joins discrete and continuous diffusion is interesting and clearly demonstrates its advantage compared to discrete diffusion.


Overall, paper is well written and easy to follow. Results are demonstrated on the large-scale SketchGraphs dataset. Related work section definitely needs more work to be done. There are also some over-claims or unsupported analysis in the paper (see my comments above).

In terms of results, I hope authors can clarify the number reported for vitruvion baseline and add the novel/unique metrics which show that model is not overfitting.

The lack of constraints in the output is also a major disadvantage of this aproach. Usually autoregressive model or pointer network are more suitable for generating the constraints. It is much more difficult for diffusion models to represent inter-relations with fixed-length parameters.

**Questions For Authors:**

1) What is the inference speed compared to autoregressive baselines?
2) Since the parameters are superpositioned, then the correct primitive type can be determined by looking at which parameters is valid after denoising, is discrete primitive type class even required?
3) How does this approach compared to other methods that jointly denoise continous and discrete values? There are many baseline diffusion methods that authors did not compared against. A very related one is House Diffusion in which discrete parameter was represented as binary hash code.

**Relation To Broader Scientific Literature:**

Automatically generating CAD engineering drawings is an interesting topic. Although this paper has somewhat limited impact in the CAD community as it removes the constraints for simplicity.

**Theoretical Claims:**

Derivation for Gaussian-softmax seems correct.

---

> ### Author Rebuttal · Authors · 2025-03-31
>
> 1. Line 108: “we propose the first diffusion-based generative model for parametric CAD sketches”.
>    1. Sorry, you are right, what we meant to write was “the first ‘sketch space’/’data space’ diffusion-based generative model for parametric CAD sketches.
> 2. No closest-retrieval results to demonstrate the model is not overfitting to the training set.
>    1. None of our baselines performed a closest-retrieval analysis to demonstrate the model is overfitting, so we saw no need to do it as well. Furthermore, we instead provided precision and recall scores since they are widely used. If need be, we can present the training and validation loss to show our model is unlikely to have overfit.
> 3. Vitruvion (table 1\) is not the same as those reported in the original paper.
>    1. Our deduplication procedure is slightly different from Vitruvion’s, where we quantize the continuous parameters to 8-bits rather than 6-bits, this was to avoid placing somewhat similar sketches into the same bin. Secondly, since our model is permutation invariant, we sorted the rows in the sketch matrix to a canonical ordering. This was done to avoid sketches with identical geometry but differing orderings from being labeled as nonidentical. As a result, our dataset differed from that used in the original Vitruvion paper, so retraining Vitruvion with our dataset yielded the scores we present in the paper. We will put this in refined paper.
> 4. Constraints are removed for simplicity, making the paper have a limited impact.
>    1. This is not as large of an issue as there are auto constraining models, for instance in Vitruvion, primitives are generated first, then constraints are filled in between them. Even though our method generates primitives, it can be coupled with “off-the-shelf” auto constraining models to create full fledged sketches. We will discuss this in the refined paper.
> 5. Data representation is similar to DeepCAD and Vitruvion without too many modifications.
>    1. The similarities between our data representation and DeepCAD/Vitruvion lies in the fact that we choose similar attributes to define the parameters of primitives. Even then, our work encodes “Arc” primitives in a different manner to DeepCAD/Vitruvion. Furthermore, Vitruvion tokenizes primitive parameters into a series of value, id, and position tokens which we don’t do. As for DeepCAD, we don’t quantize continuous parameters and we also allow constructible primitives unlike DeepCAD.
> 6. What is the inference speed compared to autoregressive baselines?
>    1. The inference speed is much slower, since we chose T \= 2000\. Our model takes \~30s to generate a sketch whereas Vitruvion takes \~3-5s. However, we believe that future work can reduce this discrepancy by perhaps borrowing from preexisting methods used in standard gaussian diffusion.
> 7. Since the parameters are superpositioned, then the correct primitive type can be determined by looking at which parameters is valid after denoising, is discrete primitive type class even required?
>    1. The primitive type is needed to determine what parameters are valid at the end of the generation, since all parameter values between \-1 and 1 are valid. Furthermore, the primitive type is needed in the reverse process/inference to downweight irrelevant parameters (Section 5.2, line 319). Without explicitly using the primitive type, because of superposition, all continuous parameters may be valid and it wouldn’t be straightforward to select one primitive type over another.
> 8. How does this approach compared to other methods that jointly denoise continous and discrete values (specifically House diffusion)
>    1. In House diffusion, discrete variables are mapped to continuous space, where the gaussian diffusion process occurs. Only gaussian diffusion is being performed, no diffusion is occurring in discrete space. For t \< 20, the model outputs a binary representation that gets mapped back to a continuous value where gaussian denoising occurs. This is not related to performing diffusion in discrete space for discrete variables and in continuous space for continuous variables concurrently.
> 9. Brepgen reference for permutation invariance missing
>    1. Thank you for letting us know, we have rectified this in our refined paper.
> 10. References missing
>     1. We have expanded on the related works section in our refined paper.
>         *Diffusion-CAD paper was published on Jan 29, 2025 which is a day before this paper was submitted.*
> 11. “For that, I am expecting to see a comparison where different primitives are separated into different tokens, with only parameters related to that primitive type included.”
>     1. I’m confused as to what you’re asking here. What difference does it make whether irrelevant parameters are zeroed out or not included? Passing either into a linear layer, will only add up the contributions of the nonzero entries. Which is exactly the case for our categorical diffusion ablation study.

---

### Official Review · Reviewer_zoeo · 2025-03-23

**Overall Recommendation:** 3

**Summary:**

This paper introduces a diffusion-based generative model for CAD sketch primitive generation. The technical innovation is a Gaussian-Softmax based diffusion paradigm. The method addresses key challenges in the heterogeneity of primitives (each primitive type is defined by its distinct parameterization) and permutation invariance of primitives. The proposed method improves the state-of-the-art results on the SketchGraphs dataset. This work marks the first application of diffusion models to parametric CAD sketch generation.

**Claims And Evidence:**

Yes

**Essential References Not Discussed:**

"Brepgen: A b-rep generative diffusion model with structured latent geometry" is also a CAD(Brep) generation paper using a diffusion-based generative model.

**Experimental Designs Or Analyses:**

yes

**Methods And Evaluation Criteria:**

yes

**Other Comments Or Suggestions:**

1. More qualitative results would be better.

**Other Strengths And Weaknesses:**

Strengths:
1. Writing is clear and easy to follow.
2. Experimental analysis is well-presented.

Weakness:
1. No qualitative comparison with other methods.
2. No failure case analysis.
3. In the ablation study,  SketchDNN (Cat.) is trained using categorical diffusion. Does this involve quantizing continuous parameters to make them discrete? This detail is unclear.
4. The number of primitives varies for each sketch. The paper does not explain how the number of primitives is determined during testing or how the data is prepared during training.

**Questions For Authors:**

See Strengths and Weaknesses.

**Relation To Broader Scientific Literature:**

The key contribution of the proposed method is its potential to address the challenges posed by heterogeneous and unordered primitives in CAD sketches.

**Theoretical Claims:**

I checked the Gaussian-Softmax Diffusion Derivation in the supplementary. There is no issue as far as I know.

---

> ### Author Rebuttal · Authors · 2025-03-31
>
> 1. No Qualitative Analysis
>    1. We have now written a qualitative analysis ready for the final paper. However, we only compare ours against Vitruvion, as other prior arts like SketchGen don’t have published code. We chose not to do a qualitative analysis against Sketchgraphs since Vitruvion is a follow up work to it.
> 2. No Failure Case Analysis
>    1. We now have a failure case analysis written and ready for the final paper. The failure cases we discuss are: 1\) The generated primitives have no discernible pattern or form, 2\) gaps exist between primitive terminations or in other words the endpoints of primitives that should be coincident are not, 3\) Extraneous primitives that don’t contribute to the overall sketch design or are not intertwined with the rest of the sketch
> 3. In the ablation study, SketchDNN (Cat.) is trained using categorical diffusion. Does this involve quantizing continuous parameters to make them discrete? This detail is unclear.
>    1. No, continuous parameters are not quantized. The only change is that the forward and reverse process of discrete variables (primitive types, constructible tag) follows the methodology set by Hoogeboom et al. in “Argmax Flows and Multinomial Diffusion: Learning Categorical Distributions” which doesn’t allow for superposition.
> 4. The number of primitives varies for each sketch. The paper does not explain how the number of primitives is determined during testing or how the data is prepared during training.
>    1. We set the maximum number of primitives to be 16, due to time and resource constraints (Section 5.1, line 305). If a sketch contains less than 16 primitives, then it is padded with null primitives which are represented by the “none” node type (Section 2, line 104). Each sketch is given by a matrix, where each row represents a primitive (Section 2, line 110).
> 5. Brepgen reference missing
>    1. Thank you for letting us know, we will include this in our final paper, as we reworked our related work section to be more comprehensive.

---

### Decision · Program_Chairs · 2025-05-01

**Decision:**

Accept (poster)

**Comment:**

Three reviews were submitted. One reviewer provided minimal feedback and was given small weight in the final decision. Among the remaining two, one gave a Weak Accept and the other a Weak Reject.

The paper introduces a continuous-discrete diffusion process (Gaussian-Softmax) for CAD sketch generation. This was recognized as a sound contribution. The rebuttal addressed concerns about discrepancies in baseline reproduction (Vitruvion) with a reasonable explanation.

The lack of constraint modeling remains a valid concern, but was not deemed strong enough on its own to warrant rejection. Overall, the contribution is clear, and the decision is accept.